# Remote epitaxy of single-crystal rhombohedral WS$_2$ bilayers

Chao Chang[1,2,7], Xiaowen Zhang[1,2,7], Weixuan Li[1,2,7], Quanlin Guo[3,7], Zuo Feng[3,7], Chen Huang[3], Yunlong Ren [1,2,4], Yingying Cai[1,2], Xu Zhou [1,2], Jinhuan Wang[1,2], Zhilie Tang[1,2], Feng Ding [5], Wenya Wei [1,2] ✉, Kaihui Liu [3,4,6] ✉ & Xiaozhi Xu [1,2] ✉

Compared to transition metal dichalcogenide (TMD) monolayers, rhombohedral-stacked (R-stacked) TMD bilayers exhibit remarkable electrical performance, enhanced nonlinear optical response, giant piezo-photovoltaic effect and intrinsic interfacial ferroelectricity. However, from a thermodynamics perspective, the formation energies of R-stacked and hexagonal-stacked (H-stacked) TMD bilayers are nearly identical, leading to mixed stacking of both H- and R-stacked bilayers in epitaxial films. Here, we report the remote epitaxy of centimetre-scale single-crystal R-stacked WS$_2$ bilayer films on sapphire substrates. The bilayer growth is realized by a high flux feeding of the tungsten source at high temperature on substrates. The R-stacked configuration is achieved by the symmetry breaking in $a$-plane sapphire, where the influence of atomic steps passes through the lower TMD layer and controls the R-stacking of the upper layer. The as-grown R-stacked bilayers show up-to-30-fold enhancements in carrier mobility (34 cm$^2$V$^{-1}$s$^{-1}$), nearly doubled circular helicity (61%) and interfacial ferroelectricity, in contrast to monolayer films. Our work reveals a growth mechanism to obtain stacking-controlled bilayer TMD single crystals, and promotes large-scale applications of R-stacked TMD.

In addition to the remarkable properties of monolayer transition metal dichalcogenide (TMD), bilayer TMD possess superior physical properties owing to the additional degrees of freedom, i.e., twist angle and stacking[1–10]. Typically, bilayer TMD exhibit two main stacking configurations: a rhombohedral-stacked (R-stacked) structure where the layers are parallelly stacked and a hexagonal-stacked (H-stacked) structure where they are antiparallelly stacked. In practice, the R-stacked bilayer TMD, which break both the in-plane and out-of-plane mirror symmetries, are highly desirable because of their exceptional optical, electrical, photovoltaic, and ferroelectric properties[11–16]. To fully realize their potential applications, large R-stacked bilayer TMD single crystals are urgently needed.

To achieve the growth of R-stacked bilayer TMD single crystals, several fundamental challenges need to be addressed: (i) reducing the

[1]Guangdong Basic Research Center of Excellence for Structure and Fundamental Interactions of Matter, Guangdong Provincial Key Laboratory of Quantum Engineering and Quantum Materials, School of Physics, South China Normal University, Guangzhou 510006, China. [2]Guangdong-Hong Kong Joint Laboratory of Quantum Matter, Frontier Research Institute for Physics, South China Normal University, Guangzhou 510006, China. [3]State Key Laboratory for Mesoscopic Physics, Frontiers Science Center for Nano-optoelectronics, School of Physics, Peking University, 100871 Beijing, China. [4]Songshan Lake Materials Laboratory, Institute of Physics, Chinese Academy of Sciences, Dongguan 523808, China. [5]Faculty of Materials Science and Engineering/Institute of Technology for Carbon Neutrality, Shenzhen Institute of Advanced Technology, Chinese Academy of Sciences, Shenzhen 518055, China. [6]Interdisciplinary Institute of Light-Element Quantum Materials and Research Centre for Light-Element Advanced Materials, Peking University, 100871 Beijing, China. [7]These authors contributed equally: Chao Chang, Xiaowen Zhang, Weixuan Li, Quanlin Guo, Zuo Feng. ✉e-mail: wywei2021@m.scnu.edu.cn; khliu@pku.edu.cn; xiaozhixu@scnu.edu.cn

nucleation barrier of bilayer TMD to ensure the preferred bilayer growth to monolayer; (ii) ensuring the simultaneous growth of both upper and lower layers of TMD to produce a uniform film; and (iii) effectively distinguishing between R- and H-stacked configurations to ensure only R-stacking in the bilayer TMD. To date, considerable efforts have been devoted to addressing the first two challenges, and polycrystalline bilayer TMD films have been reported very recently[11–16]. Nevertheless, as R- and H-stacked bilayer TMD are both thermodynamically favourable and have similar formation energies, uncontrollable stacking and grain boundaries are inevitable in the grown bilayer TMD films[13,14].

Here, we report the remote epitaxy of centimetre-scale single-crystal R-stacked bilayer $WS_2$ films on $a$-plane sapphire. Our strategy demonstrates that (i) a high W source flux at high temperature can effectively decrease the bilayer nucleation barrier, (ii) the choice of substrates with weak substrate-$WS_2$ interactions can precisely control the growth of both upper- and lower-layer $WS_2$ with nearly identical sizes, and (iii) symmetry breaking in sapphire with atomic steps can pass through the lower layer and control the R-stacking of the upper layer. The uniformly aligned R-stacked bilayer $WS_2$ islands will ultimately seamlessly stitch into a continuous single-crystal film.

## Results

### Uniform nucleation and growth of bilayer $WS_2$ islands

In principle, two approaches are typically used to obtain uniform R-stacked bilayer TMD: layer-by-layer epitaxy and simultaneous bilayer nucleation epitaxy. In the case of layer-by-layer epitaxy, due to the difficulty in achieving clean interfaces over large areas and the precise nucleation control of the upper layer, the growth of R-stacked uniform bilayer single crystals is nearly impossible[14]. In contrast, in simultaneous bilayer nucleation epitaxy, both the orientations and stacking configurations of the upper and lower layers are simultaneously determined at the early growth stage. Therefore, if the bilayer nucleation can be controlled with the same orientation, large-scale bilayer TMD single crystals is possible.

Because the nucleation barrier of bilayer TMD is usually very high, the growth of monolayer TMD on sapphire surface is generally preferred[17–25]. Therefore, new strategies need to be explored to effectively decrease the nucleation barrier (Fig. 1a). To address this issue, we conducted density functional theory (DFT) calculations and set up a thermodynamic model to investigate the nucleation and growth of bilayer $WS_2$ under different conditions (see Methods and Supplementary Note 1 for details). Since the source of S is usually present in excess during TMD growth, the concentration of W becomes a controlling factor[13]. The calculated Gibbs free energy of bilayer $WS_2$ demonstrates that the nucleation barrier can be significantly reduced from 6.75 to 1.55 eV by increasing the concentration of the W source and temperature (Fig. 1b; the corresponding $\Delta\mu_W$ values range from 0 to 0.09 eV, where $\Delta\mu_W$ represents the chemical potential difference of the W source). A change in $\Delta\mu_W$ of ~0.09 eV can be achieved by increasing the temperature for ~150 K or by increasing the partial pressure of the W source by ~30 Pa (Supplementary Fig. 1a). This change can increase the bilayer nucleation rate by ~$6 \times 10^7$ times (Supplementary Fig. 1b). In our experiment, the high partial pressure of the W source is achieved by using sufficient tungsten oxide under high temperature (see Methods for details).

Once the bilayer nucleation barrier is overcome, the thermodynamic stability of the bilayer $WS_2$ during growth is influenced by two primary factors: (i) the energy penalty upon edge formation for the upper layer TMD and (ii) the competition of the van der Waals (vdW) interaction between the substrate-TMD and TMD-TMD. The edge formation energies and vdW interactions between TMD-TMD remain constant for a specific type of TMD. Therefore, the thermodynamic stability is mainly determined by the substrate-TMD interaction. Strong interaction leads to a significantly higher energy penalty upon upper layer formation, favouring the growth of only monolayer. Therefore, the selection of a substrate with weak vdW interactions is crucial for achieving uniform bilayer TMD growth.

In line with this principle, we calculated the interfacial couplings of $WS_2$ and various sapphire planes, and found that the vdW interaction between $a$-plane sapphire and $WS_2$ is much weaker than the $WS_2$-$WS_2$ interaction, while the $c$-plane sapphire exhibits a much stronger interaction with $WS_2$ (Fig. 1c). Thus, we selected $a$-plane sapphire as the target substrate and the calculated Gibbs energy difference between monolayer and bilayer $WS_2$ demonstrated that bilayer $WS_2$ became considerably more thermodynamically favourable than the monolayer

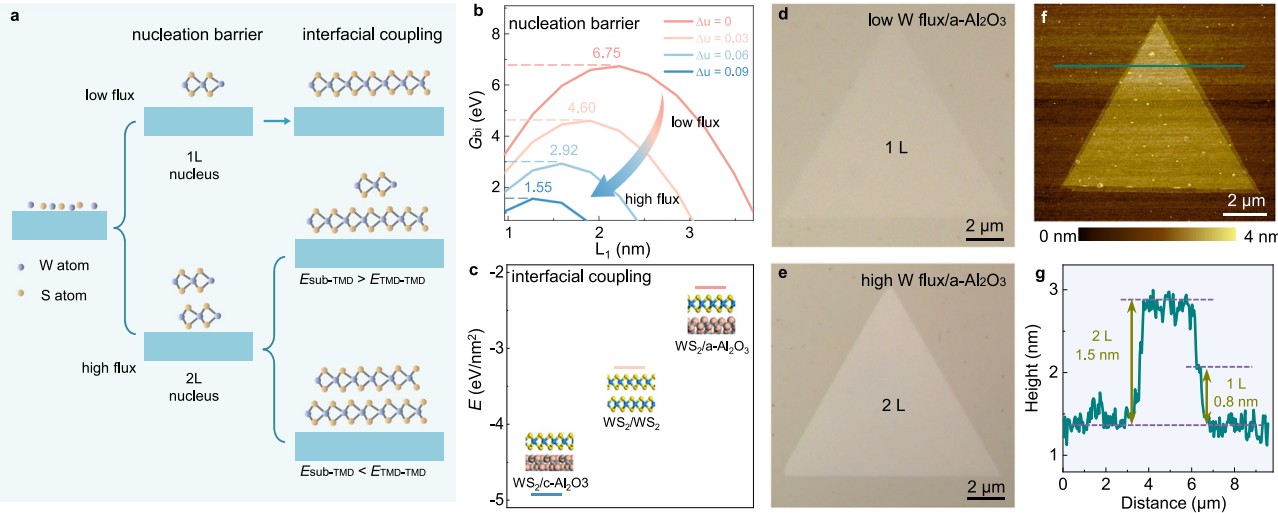

**Fig. 1 | Uniform nucleation and growth of bilayer $WS_2$ islands. a** Schematic diagram of the route to grow uniform bilayer $WS_2$. Nucleation barrier and interfacial coupling should be both considered. TMD represents the transition metal dichalcogenide, $E_{sub-TMD}$ represents the coupling between the substrate and TMD, and $E_{TMD-TMD}$ represents the coupling between two TMD layers. **b** The calculated Gibbs free energy of the bilayer $WS_2$ versus the chemical potential differences ($\Delta\mu$) of the W source, where the point with the highest G value is the nucleation barrier of bilayer $WS_2$. The nucleation barriers decrease with the increase of $\Delta\mu$ (shown with an arrow). **c** The van der Waals interaction between $WS_2$/$c$-$Al_2O_3$, $WS_2$/$WS_2$ and $WS_2$/$a$-$Al_2O_3$, respectively. Optical images of monolayer $WS_2$ (**d**) and bilayer $WS_2$ (**e**) islands obtained under low and high W flux conditions, respectively. **f** Atomic force microscopy (AFM) image of an as-grown bilayer $WS_2$ island. **g** Height profile of the bilayer $WS_2$ island in (**f**). 1 L and 2 L represent monolayer and bilayer $WS_2$, respectively.

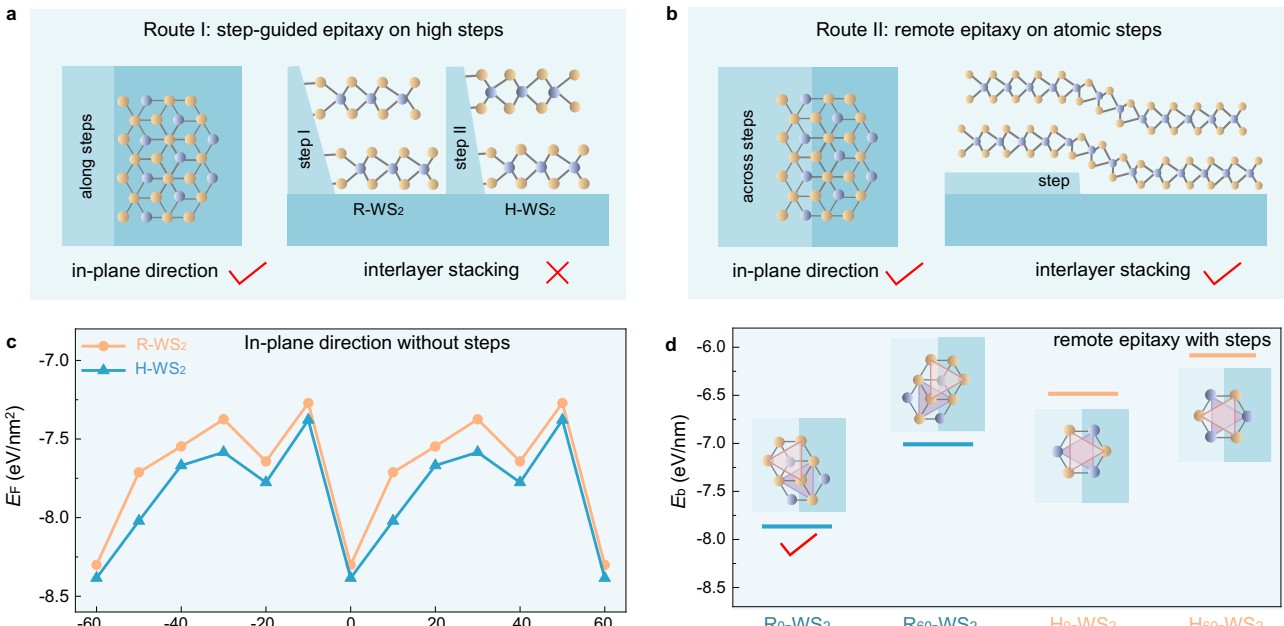

**Fig. 2 | Mechanism of remote epitaxy of R-stacked bilayer WS$_2$ single crystals on *a*-plane sapphire. a** Schematic diagrams of the step-guided epitaxy of bilayer WS$_2$ on high steps. Each layer of TMD bonds to the side of the steps. **b** Schematic diagrams of the remote epitaxy of bilayer WS$_2$ on atomic steps. The in-plane direction and interlayer stacking of bilayer WS$_2$ can both be tuned by the steps. **c** Formation energies of R- and H-stacked bilayer WS$_2$ with different rotation angles on *a*-plane sapphire without steps. Bilayer TMD has the same deep minima as monolayer ($\theta = 0°$ and $\theta = \pm60°$). **d** Binding energies of R- and H-stacked bilayer WS$_2$ that across an atomic step on *a*-plane sapphire.

WS$_2$ beyond a small critical size (~11 nm; see details in Supplementary Fig. 1c). In contrast, monolayer WS$_2$ is always preferred on *c*-plane sapphire (Supplementary Fig. 2). With this design, we successfully achieved monolayer and bilayer WS$_2$ islands with low and high W fluxes, respectively, on *a*-plane sapphire (Fig. 1d, e). Subsequent atomic force microscopy (AFM), Raman and photoluminescence (PL) characterizations further confirmed the nature of the uniform bilayer islands (Fig. 1f, g and Supplementary Fig. 3a, b).

### Remote epitaxy mechanism of R-stacked bilayer WS$_2$

In addition to achieving uniform bilayer WS$_2$ growth, precise control of the lattice orientation and stacking configuration is the key for obtaining R-stacked bilayer WS$_2$ single crystals (Fig. 2a, b). On *a*-plane sapphire, both antiparallel H- and R-stacked bilayers are thermodynamically favourable with nearly degenerate formation energies (upper plane in Fig. 2b); thus, the single-crystal growth is very challenging. Inspired by the growth of noncentrosymmetric two-dimensional (2D) monolayer hexagonal boron nitride (hBN) and TMD, we introduced parallel atomic steps on sapphire to overcome the energy equivalence of the antiparallel R- and H-stacked bilayer WS$_2$.

When these steps are involved in the growth of bilayer TMD single crystals, two mechanisms can be selected: step-guided epitaxy on high steps and remote epitaxy on atomic steps. For epitaxy with high steps, each layer of TMD bonds to the side of the steps, and the stacking configuration is strongly influenced by the atomic structure of the side surface. This step controlled process is highly complex and uncontrollable on a large scale (Fig. 2a)[26], leading to the extreme challenge in controlled stacking and single-crystal growth[13]. For remote epitaxy on atomic steps, the TMD layers are not directly bonded to the steps, and the growth mechanism is similar to the dual-coupling-guided epitaxy[27]. The TMD-TMD interaction first leads to preferred orientations and stacking configurations of the TMD, and then the remote sapphire step-TMD interaction restricts the

orientation into a single one (Fig. 2b). Evidently, the latter approach is more suitable for producing large-area bilayer TMD single crystals.

Experimentally, we observed that the morphology of the steps on a sapphire substrate can be faithfully replicated to the monolayer WS$_2$, even when the height of the steps is only ~2 Å (Supplementary Fig. 4). These results indicate that the underlying steps can tune the growth behaviours of both the lower and upper WS$_2$ layers. To further investigate this phenomenon, we conducted DFT calculations on the bilayer WS$_2$ islands on *a*-plane sapphire with atomic steps. The calculated data demonstrated that the energy equivalence of the antiparallel R$_0$-WS$_2$, R$_{60}$-WS$_2$, H$_0$-WS$_2$, and H$_{60}$-WS$_2$ are broken effectively, and only R$_0$-WS$_2$ was energetically favoured (Fig. 2c, d, the definition of R$_0$-WS$_2$, R$_{60}$-WS$_2$, H$_0$-WS$_2$, and H$_{60}$-WS$_2$ is shown in Supplementary Fig. 5). Therefore, the presence of lower layer WS$_2$ did not completely shield the potential field from the sapphire substrates, thereby enabling remote epitaxy of single-crystal bilayer WS$_2$.

### Growth of R-stacked bilayer WS$_2$

To verify our design, we conducted chemical vapour deposition (CVD) growth of WS$_2$ on *a*-plane sapphire substrates (see Methods for details). The experimental results convincingly demonstrated the successful production of single-crystal WS$_2$ films. With the assistance of atomic steps, unidirectionally aligned bilayer WS$_2$ islands could be achieved (Fig. 3a, the bilayer nucleation is shown in Supplementary Fig. 6). The aberration-corrected transmission electron microscopy (TEM) was first conducted to directly show the R-stacking lattice (Supplementary Fig. 7). Second harmonic generation (SHG) characterization was subsequently performed to identify the R-stacked bilayer WS$_2$ at large scale. Due to the non-centrosymmetric lattice of the R-stacked bilayer WS$_2$, the SHG intensity was four times greater than that of the monolayer (Fig. 3b)[28]. In contrast, the H-stacked bilayer WS$_2$ was centrosymmetric and thus exhibited a negligible SHG signal (Fig. 3b)[29]. The strong and uniform SHG intensity confirmed the R-stacked configurations of the as-grown bilayer WS$_2$ islands at a large

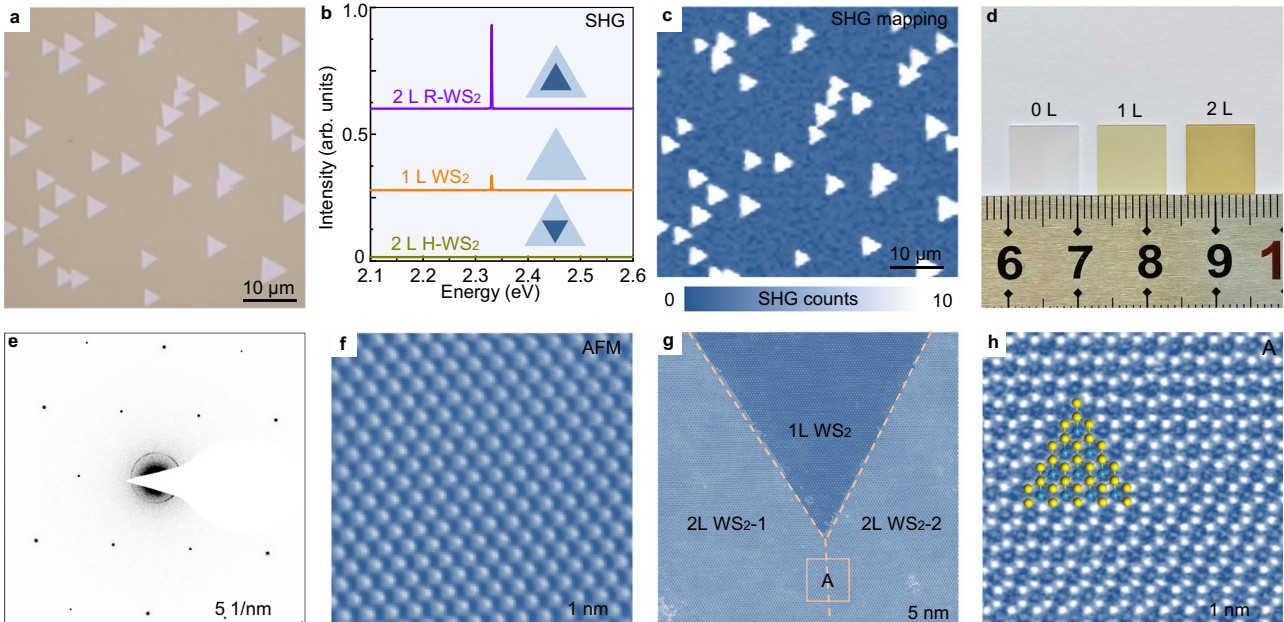

**Fig. 3 | Characterizations of R-stacked bilayer WS₂ single crystals. a** Optical image of aligned bilayer WS₂ islands. **b** Second harmonic generation (SHG) spectra of monolayer WS₂, bilayer R-WS₂ and bilayer H-WS₂ samples. **c** SHG mapping of the as-grown WS₂ samples in (**a**). **d** Optical image of the sapphire substrate, monolayer (1 L) WS₂ and bilayer (2 L) WS₂ samples. **e** Typical selected area electron diffraction (SAED) pattern of the bilayer WS₂ samples. **f** Typical AFM image of the bilayer WS₂ samples. **g** Atomically-resolved scanning transmission electron microscopy (STEM) image from the merged area of two aligned bilayer WS₂ islands (namely, 2 L WS₂-1 and 2 L WS₂-2), showing that no boundary was formed. The dashed lines represent the edges of 2 L WS₂-1 and 2 L WS₂-2. The box labelled 'A' represents a merged area. **h** Atomically-resolved STEM image of the high-quality R-stacked bilayer WS₂ lattice. The yellow and blue balls represent the W and S atoms, respectively.

scale (Fig. 3c). Continuous R-stacked bilayer WS₂ films could be obtained by increasing the growth time (Fig. 3d).

To verify the single crystallinity of the bilayer WS₂, systematic characterizations using selected area electron diffraction (SAED), polarization-dependent SHG, atomic force microscopy (AFM) and TEM were performed. The atomically-resolved TEM images and SAED patterns of 4 × 4 arrays randomly selected over the sample confirmed the alignment of the R-stacked WS₂ lattice (Fig. 3e and Supplementary Fig. 8–9). Larger-scale characterizations of polarization-dependent SHG pattern, SHG mapping and AFM images at different locations on a 1 × 1 cm² sample also confirmed the excellent alignment and absence of grain boundaries (Fig. 3f and Supplementary Fig. 10–12). Further atomic-resolved TEM images clearly verified the seamless stitching of the merged bilayer WS₂ islands (Fig. 3g, h), which was consistent with the growth of monolayer TMD[17–22]. This technique was also demonstrated applicable for the growth of aligned bilayer WSe₂ (Supplementary Fig. 13).

**Quality of R-stacked bilayer WS₂**

To evaluate the quality of the obtained R-stacked bilayer WS₂ films, electrical, optical and ferroelectric characterizations were performed. First, we fabricated conventional field-effect transistor (FET) devices of WS₂ transferred onto SiO₂/Si substrates. Remarkably to monolayer, bilayer WS₂ exhibited significantly enhanced electrical performance with ~30 times greater mobility and an ~100 times greater on/off ratio at room temperature (Fig. 4a, b, the mobility distribution of a 4 × 4 device array is shown in Supplementary Fig. 14). Experimentally, the contact capacity of monolayer WS₂ was very poor among the various TMD. Therefore, this remarkable improvement was likely attributed to the improved contact of bilayer WS₂ alongside the intrinsically higher mobility. The mobility enhancement in bilayers was also observed before in exfoliated MoS₂[13,14]. The circularly polarized PL spectra of bilayer WS₂ exhibited a much greater circular helicity than monolayer WS₂ (Fig. 4c, d), indicating promising potential applications in valley

electronics. This high value was likely attributed to the shorter exciton lifetime of bilayer WS₂[30]. Finally, we checked the ferroelectricity of R-stacked bilayer WS₂. The absence of mirror symmetry in R-stacked bilayer WS₂ induced interlayer charge transfer through hybridization between the occupied states of one layer and the unoccupied states of the other layer, generating an out-of-plane electric dipole moment and inducing interfacial ferroelectricity[31,32]. This intrinsic ferroelectricity could be observed in our bilayer WS₂ samples (Fig. 4e, f and Supplementary Fig. 15). We also tested the stability of bilayer and monolayer WS₂ samples in natural environments. After being exposed to air for two months, the monolayer WS₂ suffered obvious damage, whereas the bilayer one exhibited undetectable changes, demonstrating the superior stability of R-stacked bilayer WS₂ compared with monolayer one (Supplementary Fig. 16).

## Discussion

In conclusion, we proposed a remote epitaxy mechanism for producing R-stacked bilayer TMD single crystals. Uniform bilayer growth was achieved by introducing a high flux of W at a high temperature on substrates with weak interfacial coupling to TMD. The unidirectionally R-stacked configuration was attained by the remote symmetry breaking of the atomic steps. The as-grown single-crystal R-stacked bilayer WS₂ films exhibited significantly enhanced electrical, optical and ferroelectric properties. This mechanism, in principle, also has great potential for achieving stacking controlled few-layer TMD single crystals.

## Methods

**Growth of R-stacked bilayer WS₂ single crystals on *a*-plane sapphire**

The bilayer WS₂ films were grown on *a*-plane sapphire in a CVD system with three temperature zones, namely, zones I–III. Sulphur (1.5 g, Alfa Aesar, 99.9%) powder was placed at the upstream end of a quartz tube and heated by an extra CVD system with one temperature zone. WO₃

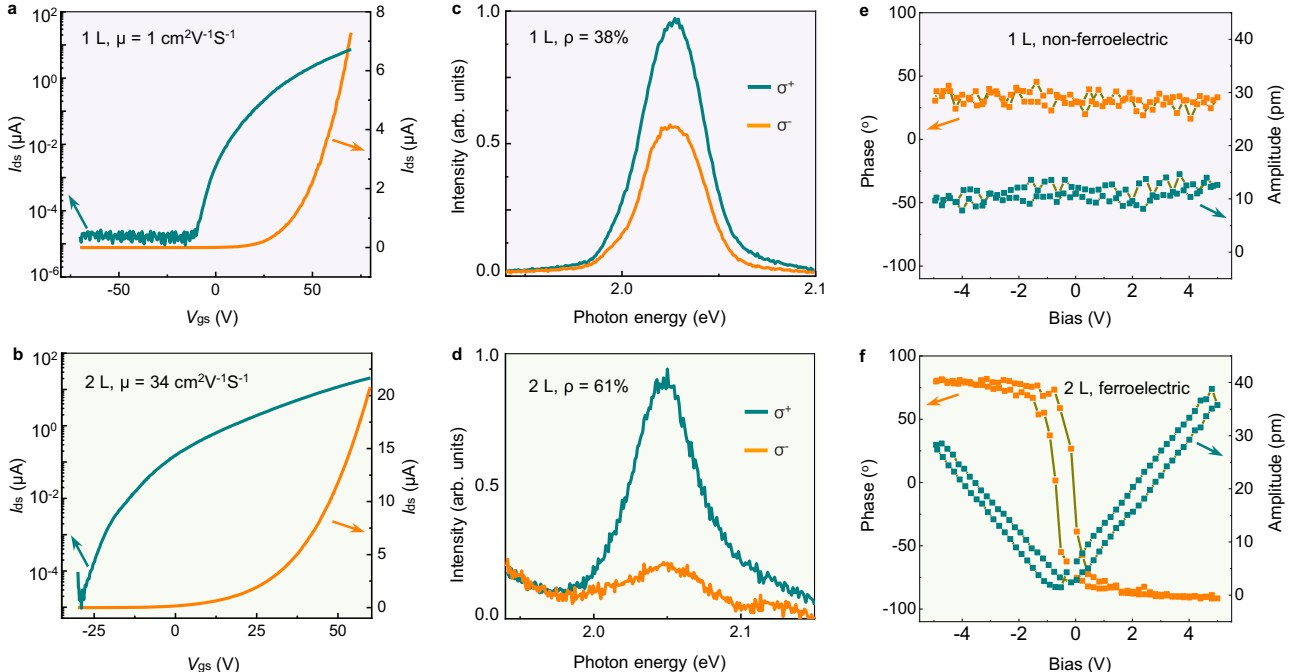

**Fig. 4 | Electrical, optical and ferroelectric properties of as-grown R-stacked bilayer WS$_2$ samples.** Electrical measurements of monolayer (**a**) and bilayer (**b**) WS$_2$ samples at room temperature, µ is the carrier mobility. Circularly polarized photoluminescence (PL) spectra of monolayer (**c**) and bilayer (**d**) WS$_2$ samples on *a*-plane sapphire, ρ is the circular helicity. The local piezo force microscopy (PFM) amplitude and phase loops during the switching process of monolayer (**e**) and bilayer (**f**) WS$_2$ samples.

(400 mg, Alfa Aesar, 99.998%) powder and NaCl (60 mg, Alfa Aesar, 99.95%) were placed in zone I of the tube furnace and sapphire substrates were placed in zone III. The NaCl can effectively lower the melting point and react with WO$_3$, resulting in much higher W source. During the growth process, the S source started to heat at 30 min and the temperature up to 160 °C within 20 min. The temperatures of zone I, zone II and zone III, were heated to 625, 625 and 975 °C in 50 min respectively, under a mixed-gas flow (Ar, 40 sccm; H$_2$, 0-1 sccm). The pressure in the growth chamber was kept at -120 Pa. After growth for 10 min, the whole CVD system was cooled down to room temperature under an Ar gas flow (40 sccm). To grow bilayer WS$_2$ islands, the quantity of WO$_3$, NaCl and S are 200 mg, 30 mg and 1 g, respectively, and the growth time is 4 min.

### Device fabrications and measurements

The FETs were fabricated through a standard microfabrication process by electron beam lithography techniques on transferred WS$_2$ on 300 nm SiO$_2$/Si. The Au contact electrodes (-50 nm) were fabricated by an e-beam deposition system with a low vacuum of -3 × 10$^{-7}$ Pa. All the electrical measurements were carried out in a probe station (base pressure 10$^{-4}$ Pa) with an Agilent semiconductor parameter analyser (B1500, high-resolution modules) at room temperature.

### Characterization

(i) AFM and PFM measurements were conducted using two types of instruments, specifically the Bruker Dimensional ICON and Asylum Cypher S. The details to obtain atomically-resolved images are as follows: The AFM measurements were conducted using an Asylum Cypher S system at room temperature under ambient condition. The system was set to lateral force microscopy mode. The setpoint was adjusted to 0.7 V and the scan rate was established at 40 Hz. The rapid scanning enabled the acquisition of the lateral signals from the samples. These signals were subsequently processed with a fast Fourier transform filter to obtain the atomically-resolved images of the samples.

(ii) Optical measurements. Optical images were conducted with an Mshot MSX10 microscope. Raman and PL spectra were conducted on a WITec-Alpha300 Raman system with a laser excitation wavelength of 532 nm and power of -2 mW. The circular helicity of the films was probed under an off-resonant excitation photon energy of 2.34 eV. Polarized light was generated with a super-achromatic quarter-wave plate (Thorlabs SAQWP05M-700) and the PL was analysed through the same quarter-wave plate as well as a linear polarizer. SHG mapping was obtained using the Raman system under excitation from a picosecond laser centred at 1064 nm with an average power of 200 mW (Rainbow 1064 OEM with pulse duration of 15 ps and repetition rate of 50 MHz).

(iii) TEM measurements. The WS$_2$ samples were transferred onto commercial holey carbon TEM grids (Zhongjingkeyi GIG-2010-3C). STEM experiments were performed in FEI Titan Themis G2 300 operated at 80 kV.

### Computational details

Geometric optimization and energy calculations of the TMD-Al$_2$O$_3$ systems were carried out using density functional theory (DFT) as implemented in Vienna Ab-initio Simulation Package[33,34]. The exchange-correlation functions are treated by the generalized gradient approximation[35] and the interaction between valence electrons and ion cores is carried out by the projected augmented wave method[36]. The plane-wave cutoff energy was set at 400 eV for TMD-Al$_2$O$_3$ systems. The dispersion-corrected DFT-D3 method was used because of its good description of long-range vdW interactions for multi-layered 2D materials. The geometries of the TMD-Al$_2$O$_3$ systems were relaxed until the force on each atom was less than 0.02 eV Å$^{-1}$, and the energy convergence criterion of 1 × 10$^{-4}$ eV was met. The Al$_2$O$_3$ surfaces were modelled by a periodic slab and some bottom layers were fixed to mimic the bulk, a 1 × 1 × 1 Monkhorst–Pack k-point mesh was adopted.

## Data availability

The Source Data underlying the figures of this study are available with the paper. All raw data generated during the current study are available from the corresponding authors upon request. Source data are provided with this paper.

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

## Acknowledgements

This work was supported by the National Key R&D Program of China (2022YFA1403503 (X.X.)), the National Natural Science Foundation of China (12322406 (X.X.), 52102043 (X.X.), 52025023 (K.L.), 51991342 (K.L.), 52021006 (K.L.), 52372046 (X.Z.) and 52102044 (X.Z.)), Guangdong Major Project of Basic and Applied Basic Research (2021B0301030002 (K.L.)), the Key R&D Program of Guangdong Province (2020B010189001 (X.X.), 2019B010931001 (K.L.), 2018B010109009 (D.Y.) and 2018B030327001 (D.Y.)), the Pearl River Talent Recruitment Program of Guangdong Province (2019ZT08C321 (X.X.)), the National Postdoctoral Program for Innovative Talents (BX20220117 (W.W.)), China Postdoctoral Science Foundation (2022M721224 (W.W.) and 2022M720208 (J.W.)), the Key Project of Science and Technology of Guangzhou (202201010383 (X.Z.)), Guangdong Basic and Applied Basic Research Foundation (2023A1515012743 (X.Z.)) and the Strategic Priority Research Program of Chinese Academy of Sciences (XDB33000000 (K.L.)). We thank the National Supercomputer Centre in Tianjin for computing support.

## Author contributions

X.X. and K.L. supervised the project. C.C., X.Z., W.L., Y.C. and J.W. conducted the sample growth, Q.G. performed the TEM experiments. C.H. performed the optical measurements. Y.R. performed the AFM experiments. Z.F. performed the electrical measurements. W.W. and F.D. performed the theoretical calculations. X.X., K.L. and W.W. wrote the article, Z.T., X.Z. and F.D. revised the manuscript. All of the authors discussed the results and comments on the paper.

## Competing interests

The authors declare no competing interests.
