## [Peer Review File · Nature Communications]

Remote epitaxy of single-crystal rhombohedral WS₂ bilayersEditorial Note: Parts of this Peer Review File have been redacted as indicated to remove third-party material where no permission to publish could be obtained.

REVIEWER COMMENTS

Reviewer #1 (Remarks to the Author):

The authors reported the interesting work and successfully growth of centimeter-scale single-crystal R-stacked WS₂ bilayer films on sapphire substrates. The growth mechanism has been thoroughly discussed and the sample realized good physical properties due to its high quality. It is inspiring for 2D material growth. This work has broad interest and is justify to publish in Nature communication. However, before that, the following point should be addressed.

1 This method seems have strong correlation with the subtract. Is the WS₂-sapphire interaction covalent bonding or weak van der Waals bonding?

2 Is this method a universal approach to obtain large scale TMD thin films?

3 More should be addressed on the uniqueness of R-stacked bilayer WS₂ compared with single layer WS₂ in application.

4 The color bar should be added in Figure 3c to identity to SHG Intensity.

5 Typo in figure 4f, bise change to bias.

6 The authors performed electrical measurement to show the high quality of grown bilayer WS₂. How is the uniformity of electric properties in the large scale of WS₂. The electrodes on the film is cross grain boundary or within a crystalline domain. The channel length of the transistor?. Device arrays measurement is needed.

7 Can the authors provide the ferroelectric domain structure or E-reverse ferroelectric domain PFM image.

Reviewer #2 (Remarks to the Author):

The authors of manuscript "Remote epitaxy of single-crystal rhombohedral WS₂ bilayers", proposed an epitaxy mechanism for producing rhombohedral bilayer TMDs. The growth of R-stacked bilayer WS₂ benefits from two aspects: (i) a high W source flux at high temperature can effectively decrease the bilayer nucleation barrier; (ii) the symmetry breaking in a-sapphire with atomic steps can pass through the lower layer and control the R-stacking of the upper layer. This work could provide a significant method to achieve large-sized R-stacked TMD bilayer films, which holds great importance for their potential application in advanced electronic devices.

I would like to suggest that the manuscript may be revised carefully according to the following questions:

1. Without being guided by high steps, how to achieve bilayer nucleation on substrate with atomic steps? Please provide direct experimental evidence for the bilayer nucleation.
2. To verify the single crystallinity of the bilayer WS₂, polarization-dependent SHG were performed. The authors are advised to provide TEM high-resolution atomic images and corresponding SAED images of arrays (for example 4*4) randomly selected over the entire sample.
3. It is more important that authors are advised to provide TEM high-resolution atomic images of the sample in cross-section in order to show the R-stacking.
4. Please provide detailed growth conditions for unidirectional bilayer triangulars and continuous films in the growth method.
5. In page 6, "R0-WS₂, R60-WS₂, H0-WS₂, and H60-WS₂...", what do these symbols represent?

Reviewer #3 (Remarks to the Author):

This paper focuses on the single-crystal rhombohedral WS₂ growth on a-plane sapphire, particularly pointing out the mechanism of remote epitaxy interaction through step edges in the crystal orientation and the R-stack WS₂ second layer. The findings are somewhat encouraging and may stimulate additional directions to bilayer TMDs synthesis. However, there are some technical incoherency and insufficient experimental proofs needed to be further addressed.

1. There are typos and loose usage of the terms which required more attention. For example, in Result section, paragraph 2 - the "grwoth" of monolayer TMDs on sapphire surface is generally preferred. Furthermore, the abbreviation of transition metal dichalcogenide(s) should be consistent. For example, in Result section, paragraph 3 - ... substrate-TMD and TMD-TMD... vdW interactions between TMD-TMDs remain constant ..., the authors should have proof-reading more carefully.
2. There is a report showing that bilayer-MoS₂ crystals can be directly synthesized on c-plane sapphire (Nature 605, 69-75 (2022)), which conflicts with the statement in this paper (Fig. 1c). The authors should elaborate more to further clarify the the proposed mechanism.
3. The authors claimed that "a high W source flux at high temperature" is one of the factors to achieve uniform R-stack bilayer WS₂; however, the growth substrate, method and growth temperature in this report (a-plane sapphire with 2 angstrom atomic steps/ WO₃ (400 mg, Alfa Aesar, 99.998%) powder and NaCl (60 mg, Alfa Aesar, 99.95%) were placed in zone I/ 160, 625, 625 and 975°C/ under a mixed-gas flow (Ar, 40 sccm; H₂, 0-1 sccm)/ ~120 Pa) is almost the same as the method in the previous report (Nature Nanotechnology 17, 33-38 (2022)) (a-plane sapphire with 2 angstrom atomic steps/ WO₃ (Alfa Aesar, 99.99%) powder and NaCl (Greagent, 99.95%) were placed in zone I/ 125, 645, 850 and 965°C/ Ar, 200

s.c.c.m.; H₂, 0-5 s.c.c.m./ ~300 Pa), which demonstrated monolayer growth. Based on these conditions, the temperature of the sulfur in this paper, 160°C at ~120 Pa, is much higher than the previous report, 125°C at ~300 Pa, and the temperature for WO₃ in this report is 625°C, which is lower than the previous report, 645°C, referring to a higher S to W ratio in this paper. It is necessary to investigate these details further to confirm the theory.

4. The authors claim to demonstrate the single-crystal growth of R-WS₂ bilayers and coalesce to a grain-boundary free single-crystal film with centimeter scale; however, the statement "The uniformly aligned R-stacked bilayer WS₂ islands will ultimately seamlessly stitch into a continuous single-crystal film." is not supported by experimental data, a large-scale characterization, such as SHG mapping, was omitted.

5. The data which supports the remote epitaxy interaction through steps on sapphire substrate is vague (it is common to observe steps replicated from underlying substrate on the epi-crystals, but it doesn't mean they all have remote epitaxy relation), and the data supporting the success of bilayer crystals is also not sufficient. Cross-sectional TEM may be necessary.

6. In Method section, the authors need to elaborate more on the condition and measurement details of how to get atomic-resolution AFM images in Fig. 3f and Supplementary Fig. 7 with "Bruker Dimensional ICON under an atmospheric environment."

Reply to Referee #1

Original comment (1):

The authors reported the interesting work and successfully growth of centimeter-scale single-crystal R-stacked WS₂ bilayer films on sapphire substrates. The growth mechanism has been thoroughly discussed and the sample realized good physical properties due to its high quality. It is inspiring for 2D material growth. This work has broad interest and is justify to publish in Nature communication. However, before that, the following point should be addressed.

Our reply:

We thank the referee's efforts and time in reviewing our manuscript. His/her valuable suggestions and comments are quite helpful for us to improve the quality of this work. We have added more experimental and theoretical data to address these concerns as in the below replies.

Original comment (2):

1. This method seems have strong correlation with the substrate. Is the WS₂-sapphire interaction covalent bonding or weak van der Waals bonding?

Our reply:

We thank the referee for this insightful concern on the interaction strength between WS₂ and sapphire substrates. Our method has strong correlation with the substrate. Weak substrate-WS₂ interaction enables precise control over the growth of both upper- and lower-layer WS₂ with nearly identical sizes. Conversely, strong interactions pose challenges in achieving comparable growth rates for upper layer WS₂ compared with the lower layer, leading to limited bilayer coverage on a monolayer WS₂ island.

Our calculations show that the interaction between WS₂ and *a*-plane sapphire and is ~0.06 eV/atom (5.79 kJ/mol/atom), the interaction between WS₂ and *c*-plane sapphire is ~0.14 eV/atom (13.51 kJ/mol/atom). Both are in the van der Waals bonding energy range (~10 kJ/mol), and much lower than the covalent bonding energy (~500 kJ/mol, Fig. R1, Functional Materials: Preparation, Processing and Applications, 2012, 467).

[REDACTED]

Fig. R1. Interatomic bonding energy for different interatomic bonds.

Original comment (3):

2. Is this method a universal approach to obtain large scale TMD thin films?

Our reply:

We thank the referee for the concern on the universality of our method. In principle, our method should be applicable to other TMD as well. As an example, we conducted experiments on the growth of other TMD materials and successfully achieved the uniform aligned growth of bilayer WSe₂ (Fig. R2a-b). Subsequently, we also checked this growth behaviour by density functional theory (DFT) calculations. The results further confirmed that bilayer WSe₂ can also exhibit identical orientations on the *a*-plane sapphire substrate (Fig. R2c).

We have added these data and discussions in the revised manuscript.

Fig. R2. (a) Optical image of aligned bilayer WSe₂ islands. (b) Raman spectrum of bilayer WSe₂. (c) Binding energies of R- and H-stacked bilayer WSe₂ that across an atomic step on *a*-plane sapphire

Original comment (4):

3. More should be addressed on the uniqueness of R-stacked bilayer WS₂ compared with single layer WS₂ in application.

Our reply:

We greatly thank the referee for this kind suggestion on giving more examples of the uniqueness of bilayer WS₂ compared with single-layer WS₂ in application. Following the referee's advice, we have done additional exploration on the stability of bilayer and monolayer WS₂ samples in natural environment. We systematically adjusted the growth parameters, deliberately fabricated islands with single layer and bilayer areas, and left them exposed to air for two months. The optical images demonstrate that the single-layer region suffered obvious damage, whereas the bilayer areas exhibited no detectable changes (Fig. R3). This observation clearly highlights the superior stability of bilayer WS₂ compared with single-layer WS₂, thereby bearing significant implications for practical applications. The greater stability of bilayer TMD may be attributed to their indirect band gaps that suppress the photo-oxidation processes that have been reported to be responsible for monolayer TMD degradation (*Advanced Materials Interfaces* 2021, 8, 2101188; *Nano Letters* 2019, 19, 5205).

Fig. R3. Optical image of an R-stacked bilayer WS₂ island with monolayer areas after being exposed to air for two months.

Original comment (5):

4. The colour bar should be added in Figure 3c to identify to SHG Intensity.

Our reply:

We thank the referee for this suggestion. Following the referee's advice, we have added the colour bar of Fig. 3c in the revised manuscript.

Original comment (6):

5. Typo in figure 4f, bise change to bias.

Our reply:

We greatly thank the referee for pointing out this typo. We have corrected "bise" to "bias" in the revised manuscript. In order to make the language more accurate, we applied the Nature springer language editing service and carefully proofread the entire manuscript.

Original comment (7):

6. The authors performed electrical measurement to show the high quality of grown bilayer WS₂. How is the uniformity of electric properties in the large scale of WS₂. The electrodes on the film are cross grain boundary or within a crystalline domain. The channel length of the transistor? Device arrays measurement is needed.

Our reply:

We thank the referee for raising the question on the electrical measurements of bilayer WS₂ samples. In the previous version, random device testing was conducted directly on bilayer WS₂ films, so we have no information on whether the electrodes were within a single domain or across different domains. The channel length of the transistor was 6.7 μm. To validate the uniformity of electrical properties, we followed the referee's suggestion and fabricated a 4×4 device array on the SiO₂/Si substrate (Fig. R4a). The averaged carrier mobility of the 16 devices is ~33 cm²V⁻¹S⁻¹ and the mobility distribution shown in Fig. R4b verifies the good uniformity of electrical properties.

We have added this data and discussions in the revised manuscript.

Fig. R4. (a) Optical image of a 4×4 device array. (b) Room-temperature mobility of devices shown in (a).

Original comment (8):

7. Can the authors provide the ferroelectric domain structure or E-reverse ferroelectric domain PFM image?

Our reply:

We appreciate the referee’s suggestion on providing the ferroelectric domain structure of as-grown bilayer WS₂ samples. Following the referee’s advice, we have conducted the PFM measurement. A DC tip bias of +4 V was utilized to scan the central region. This results in the formation of a domain walls that exhibit contrast in the amplitude image, effectively demonstrating the electrical switching behaviour of ferroelectric domains (Fig. R5).

We have added this data and discussions in the revised manuscript.

Fig. R5. The amplitude images of bilayer WS₂ recorded after polarization switching with DC bias +4 V in the central region.

In summary, we are very grateful to the referee's efforts in reviewing our manuscript. Especially these valuable comments and suggestions really help us to improve the manuscript significantly. We hope that our reply has fully addressed all raised concerns and the referee will enjoy the story in the revised version.

Reply to Referee #2

Original comment (1):

The authors of manuscript "Remote epitaxy of single-crystal rhombohedral WS₂ bilayers", proposed an epitaxy mechanism for producing rhombohedral bilayer TMDs. The growth of R-stacked bilayer WS₂ benefits from two aspects: (i) a high W source flux at high temperature can effectively decrease the bilayer nucleation barrier; (ii) the symmetry breaking in a-sapphire with atomic steps can pass through the lower layer and control the R-stacking of the upper layer. This work could provide a significant method to achieve large-sized R-stacked TMD bilayer films, which holds great importance for their potential application in advanced electronic devices. I would like to suggest that the manuscript may be revised carefully according to the following questions:

Our reply:

We sincerely thank the referee's efforts and time in reviewing our manuscript. These raised valuable suggestions and comments are quite helpful for us to improve the quality of this work. We have added more experimental data and discussions to address these concerns as in the below replies.

Original comment (2):

Without being guided by high steps, how to achieve bilayer nucleation on substrate with atomic steps? Please provide direct experimental evidence for the bilayer nucleation.

Our reply:

We greatly thank the referee for raising the question on the bilayer nucleation.

As the nucleation barrier of bilayer TMD is usually very high, the growth of monolayer TMD on sapphire surface is generally preferred. Therefore, the key to grow bilayer TMD is to decrease the nucleation barrier. In 2022, Wang et al. reported that the existence of high steps can help to decrease this barrier (Nature 2022, 605, 69). In this work, our calculations show that the nucleation barrier of bilayer WS₂ can also be significantly reduced from 6.75 to 1.55 eV by increasing the concentration of the W source and temperature (Fig. R1a). Therefore, we used sufficient tungsten oxide at high temperature to ensure a high W source flux, and achieve bilayer nucleation.

To provide direct experimental evidence for the bilayer nucleation, we conducted AFM measurements at the early growth stage (Fig. R1b). The AFM characterizations demonstrated the bilayer nucleation of WS₂ (Fig. R1c).

Fig. R1. (a) The calculated Gibbs free energy of the bilayer WS₂ versus the chemical potential differences of the W source, where the point with the highest G value is the nucleation barrier of bilayer WS₂. (b) AFM image of two WS₂ islands at early growth stage. (c) Height profile of (b), indicating the bilayer WS₂ nucleation.

Original comment (3):

2. To verify the single crystallinity of the bilayer WS₂, polarization-dependent SHG were performed. The authors are advised to provide TEM high-resolution atomic images and corresponding SAED images of arrays (for example 4*4) randomly selected over the entire sample.

Our reply:

We greatly appreciate the referee's suggestion on the TEM and SAED measurements. Following the referee's advice, we have conducted more TEM measurements on the bilayers. The atomically-resolved TEM images and SAED patterns of 4×4 arrays randomly selected over the sample were shown in Fig. R2-3. The data helped us to verify the single crystallinity of the bilayer WS₂.

We have added these data and discussions in the revised manuscript.

Fig. R2. (a-p) Atomically-resolved TEM images of the bilayer WS₂, the lattice orientations of WS₂ at different areas are nearly the same.

Fig. R3. (a-p) SAED patterns of the bilayer WS₂ samples at different areas of the sample, demonstrating nearly identical orientations.

Original comment (4):

3. It is more important that authors are advised to provide TEM high-resolution atomic images of the sample in cross-section in order to show the R-stacking.

Our reply:

We greatly appreciate the referee for the suggestion on the cross-section TEM measurements of the bilayer WS_2 samples. Following the referee's kind advice, we have conducted additional experiments and obtained atomically-resolved cross-section TEM images (Fig. R4). The obtained lattice confirmed the bilayer R-stacking structure.

Fig. R4. Cross-sectional TEM image of bilayer WS_2 samples.

Original comment (5):

4. Please provide detailed growth conditions for unidirectional bilayer triangles and continuous films in the growth method.

Our reply:

We thank the referee for this suggestion on providing detailed growth conditions. The growth conditions for unidirectional bilayer triangles and continuous films are very similar. The key difference lies in the quantity of the source and the growth time. Specifically, the synthesis of bilayer triangles requires less source and a shorter time compared to the production of continuous films.

For a typical growth process of bilayer triangles, sulfur (1 g, Alfa Aesar, 99.9%) powder was placed at the upstream end of a quartz tube and heated by an extra CVD system with one temperature zone. WO_3 (200 mg, Alfa Aesar, 99.998%) powder and NaCl (30 mg, Alfa Aesar, 99.95%) were placed in zone I of the tube furnace and sapphire substrates were placed in zone III. During the growth process, the S source started heating at 30 min, reaching a temperature up of 160 °C within 20 min. The temperatures of zone I, zone II and zone III, were heated raised to 625, 625 and 975 °C in 50 min under a mixed-gas flow (Ar, 40 sccm; H_2 , 0-1 sccm). The pressure in the growth chamber was kept at ~120 Pa. After the growth for 4 min, the whole CVD system was cooled down to room

temperature under an Ar gas flow (40 sccm). For a typical growth process of continuous films, the quantity of WO_3 and NaCl are doubled and the quantity of S is increased by 1.5 times, and the growth time increases from 4 min to 10 min.

We have updated these growth conditions in the method section of the revised manuscript.

Original comment (6):

5. In page 6, “ $\text{R}_0\text{-WS}_2$, $\text{R}_{60}\text{-WS}_2$, $\text{H}_0\text{-WS}_2$, and $\text{H}_{60}\text{-WS}_2$...””, what do these symbols represent?

Our reply:

We greatly thank the referee for raising this question on the definition of the symbols. Typically, bilayer TMD exhibit two main stacking configurations: a rhombohedral-stacked (R-stacked) structure where the layers are parallel stacked and a hexagonal-stacked (H-stacked) structure where they are antiparallel stacked. For bilayer WS_2 grown on the ideal surface of a-plane sapphire, there are two minimum energy points with twist angle of $\theta = 0^\circ$ and 60° , where θ is the angle between one zigzag direction of lower layer WS_2 and the $\langle 1-100 \rangle$ plane of the a-plane sapphire (Fig. 2c). We defined the bilayer R-stacked WS_2 with $\theta = 0^\circ$ as $\text{R}_0\text{-WS}_2$ and bilayer R-stacked WS_2 with $\theta = 60^\circ$ as $\text{R}_{60}\text{-WS}_2$. Using the same method, we defined the bilayer H-stacked WS_2 with $\theta = 0^\circ$ as $\text{H}_0\text{-WS}_2$ and bilayer H-stacked WS_2 with $\theta = 60^\circ$ as $\text{H}_{60}\text{-WS}_2$. In order to make this definition clearer, we added the detailed structure in the revised Supplementary information (Fig. R5).

Fig. R5. Schematic diagrams of the definition of $\text{R}_0\text{-WS}_2$, $\text{R}_{60}\text{-WS}_2$, $\text{H}_0\text{-WS}_2$, and $\text{H}_{60}\text{-WS}_2$.

In summary, we are very much grateful to the referee’s efforts in reviewing our manuscript. Especially these valuable comments and suggestions greatly help us to improve the manuscript significantly. We hope that our reply has fully addressed all raised concerns and the referee will enjoy the revised version.

Reply to Referee #3

Original comment (1):

This paper focuses on the single-crystal rhombohedral WS₂ growth on a-plane sapphire, particularly pointing out the mechanism of remote epitaxy interaction through step edges in the crystal orientation and the R-stack WS₂ second layer. The findings are somewhat encouraging and may stimulate additional directions to bilayer TMDs synthesis. However, there are some technical incoherency and insufficient experimental proofs needed to be further addressed.

Our reply:

We thank the referee's efforts and time in reviewing our manuscript. His/her valuable suggestions and comments are quite helpful for us to improve the quality of this work. We have added more experimental data and discussions to address these concerns as in the below replies.

Original comment (2):

There are typos and loose usage of the terms which required more attention. For example, in Result section, paragraph 2 - the "grwoth" of monolayer TMDs on sapphire surface is generally preferred. Furthermore, the abbreviation of transition metal dichalcogenide(s) should be consistent. For example, in Result section, paragraph 3 - ... substrate-TMD and TMD-TMD... vdW interactions between TMD-TMDs remain constant ..., the authors should have proof-reading more carefully.

Our reply:

We greatly thank the referee for pointing out these typos. We have corrected the typos in the revised manuscript. In order to make the language more accurate, we applied the Nature springer language editing service and carefully proofread the entire manuscript.

Original comment (3):

2. There is a report showing that bilayer-MoS₂ crystals can be directly synthesized on c-plane sapphire (Nature 605, 69-75 (2022)), which conflicts with the statement in this paper (Fig. 1c). The authors should elaborate more to further clarify the proposed mechanism.

Our reply:

We greatly thank the referee for raising the comparison to the previous work of bilayer TMD growth on *c*-plane sapphire.

The mentioned work by Liu et al. is a pioneering one in bilayer TMD growth (Nature 2022, 605, 69). They employed a high step on *c*-plane sapphire to reduce the nucleation barrier of the bilayer TMD, enabling simultaneous bonding of upper and lower TMD layers to ensure uniform bilayer (Fig. R1a-b). However, the growth behaviour on substrates with high steps in previous work is very different from that on substrates with atomic steps in our work.

In their work, they also mentioned that on a *c*-plane sapphire substrate with atomic steps, only monolayer MoS₂ can be grown (Fig. R1c-d), which is consistent with our calculations and experimental results.

We have added these discussions in the revised manuscript.

[REDACTED]

Fig. R1. Bilayer and monolayer MoS₂ growth on high steps and atomic steps in the reference (Nature 2022, 605, 69). [REDACTED]

Original comment (4):

3. The authors claimed that "a high W source flux at high temperature" is one of the factors to achieve uniform R-stack bilayer WS₂; however, the growth substrate, method and growth temperature in this report (a-plane sapphire with 2 angstrom atomic steps/ WO₃ (400 mg, Alfa Aesar, 99.998%) powder and NaCl (60 mg, Alfa Aesar, 99.95%) were placed in zone I/ 160, 625, 625 and 975°C/ under a mixed-gas flow (Ar, 40 sccm; H₂, 0-1 sccm)/ ~120 Pa) is almost the same as the method in the previous report (Nature Nanotechnology 17, 33-38 (2022)) (a-plane sapphire with 2 angstrom atomic steps/ WO₃ (Alfa Aesar, 99.99%) powder and NaCl (Greagent, 99.95%) were placed in zone I/ 125, 645, 850 and 965°C/ Ar, 200 s.c.c.m.; H₂, 0-5 s.c.c.m/ ~300 Pa), which demonstrated monolayer growth. Based on these conditions, the temperature of the sulfur in this paper, 160°C at ~120 Pa, is much higher than the previous report, 125°C at ~300 Pa, and the temperature for WO₃ in this report is 625°C, which is lower than the previous report, 645°C, referring to a higher S to W ratio in this paper. It is necessary to investigate these details further to confirm the theory.

Our reply:

We greatly appreciate the referee's suggestion on comparing the growth parameters of our manuscript comparing with our earlier work. We agree with the referee that the overall growth parameters were very similar to the previous one, and the temperature we used in this work was even a little bit lower than before.

The key for bilayer nucleation, as stated in the manuscript, lies in the abundant concentration of W sources. Therefore, our primary focus in adjusting experimental parameters to obtain a substantial concentration of W sources. In our experiment, we used a high amount of WO₃ (400 mg) and NaCl (60 mg). The NaCl can effectively lower the melting point and react with WO₃, resulting in much higher W source at even lower temperature (Nature 2018, 556, 355; ACS Nano 2019, 13, 3649). As a contrast, we used 20 mg WO₃ and 2 mg NaCl in the work of Nature Nanotechnology 2022, 17, 33, and can only obtain monolayer WS₂ samples.

We have added these discussions in the revised manuscript.

Original comment (5):

4. The authors claim to demonstrate the single-crystal growth of R-WS₂ bilayers and coalesce to a grain-boundary free single-crystal film with centimeter scale; however, the statement "The uniformly aligned R-stacked bilayer WS₂ islands will ultimately seamlessly stitch into a continuous single-crystal film." is not supported by experimental data, a large-scale characterization, such as SHG mapping, was omitted.

Our reply:

We greatly thank the referee for the suggestion on large-scale SHG mapping. Following the referee's advice, we have conducted SHG measurements at different areas of the samples at large scale (Fig. R2). The uniform contrast without dark lines (corresponding to grain boundaries) confirmed the seamless stitching of the WS₂ samples.

We have added the data in the revised manuscript.

Fig. R2. SHG mapping of WS₂ samples at different areas. No dark lines can be observed, demonstrating the seamless stitching of different WS₂ islands. All the images are of the same size.

Original comment (6):

5. The data which supports the remote epitaxy interaction through steps on sapphire substrate is vague (it is common to observe steps replicated from underlying substrate on the epi-crystals, but it doesn't mean they all have remote epitaxy relation), and the data supporting the success of bilayer

crystals is also not sufficient. Cross-sectional TEM may be necessary.

Our reply:

We greatly appreciate the referee's concern on the remote epitaxy interaction and the cross-sectional TEM measurements.

For the R- and H-stacked bilayer WS₂ on ideal a-plane sapphire, the binding energies are very similar (Fig. 2c). But on a-plane sapphire with steps, the binding energies of the two stacked lattices are distinct different (Fig. 2d). So, the steps on a-plane sapphire substrate can obviously tune the growth behaviours of both the lower and upper WS₂ layers. We agree with the referee that it is common to observe steps replicated from underlying substrate. So, the nucleation site on the steps edges is very important in our technique, otherwise the steps cannot work in the lattice orientation and stacking control.

In our experiment, we used un-annealed sapphire substrates to guarantee the bilayer WS₂ nucleation at step edges by accurate time sequence control of the simultaneous formation of grain nuclei and substrate steps (Nature Communications 2023, 14, 592). To provide direct experimental evidence for the bilayer nucleation at step edges, we conducted AFM measurements at the early growth stage. The AFM characterizations demonstrated the bilayer nucleation of WS₂ near step edges (Fig. R3).

Fig. R3. (a) AFM images of two WS₂ islands at early growth stage. (b) Height profile of the area shown in (b), indicating the bilayer WS₂ nucleation. (c) Schematic diagrams showing the nucleation centre near step edges.

We have also followed the referee's advice and conducted additional TEM experiments and obtained atomically-resolved cross-section TEM images and demonstrated the bilayer lattice (Fig. R4).

Fig. R4. Cross-sectional TEM image of bilayer WS₂ samples.

We have added these data and discussions in the revised manuscript.

Original comment (7):

6. In Method section, the authors need to elaborate more on the condition and measurement details of how to get atomic-resolution AFM images in Fig. 3f and Supplementary Fig. 7 with "Bruker Dimensional ICON under an atmospheric environment."

Our reply:

We greatly thank the referee for the suggestion on the condition and measurement details of atomic-resolution AFM images.

After verification with the analysis and testing centre, we found that our AFM measurements were conducted using two types of instruments, specifically the Bruker Dimensional ICON and Asylum Cypher S. The AFM images of Fig. 1f and Supplementary Fig. 4a are obtained on a normal Bruker Dimensional ICON AFM system, and the AFM images in Fig. 3f and Supplementary Fig. 7 were carried out using an Asylum Cypher S AFM system to obtain the atomic resolution.

The details to obtain Fig. 3f are as follows. The atomically-resolved AFM measurements were conducted using an Asylum Cypher S system at room temperature under ambient condition. The system was set to lateral force microscopy mode. The setpoint was adjusted to 0.7 V and the scan rate was established at 40 Hz. The rapid scanning enabled the acquisition of the lateral signals from the samples. These signals were subsequently processed with a fast Fourier transform filter to obtain the atomically-resolved images of the samples.

We have updated the above details in the revised manuscript.

In summary, we are very grateful to the referee's efforts and expertise in reviewing our manuscript. Especially these insightful comments and suggestions greatly help us to improve the manuscript significantly. We hope that our reply has fully addressed all raised concerns and the referee will enjoy the revised version.

REVIEWERS' COMMENTS

Reviewer #1 (Remarks to the Author):

The authors have done great effort to improve the manuscript and all my concerns have been addressed. At this stage, the revised manuscript should be accepted.

Reviewer #2 (Remarks to the Author):

Authors have revised the manuscript according to the comments presented by two reviewers and replied all my questions. I would like to suggest that the manuscript may be accepted after comparing the growth mechanism in this manuscript with that of review article on *Nanoscale* (2024, 16, 978 Large-area single-crystal TMD growth modulated by sapphire substrates).

Reviewer #3 (Remarks to the Author):

Many thanks to the authors for the response.

My comments to authors' response,

(2) No further comment.

(3) No further comment.

(4) No Further comment.

(5) The SHG mapping images (Supplementary Fig. 11) should include intensity scale bar, and 3R stacking should have stronger SHG intensity, but from those images it's hard to tell; otherwise, those images seem like a monolayer, noise or not on focus.

(6) The orientation of the triangular WS₂ in Supplementary Fig. 4 (with the edge of the triangle parallel to the step edge) is different from the triangle in Supplementary Fig. 6 (the edge of the triangle perpendicular to the step edge), which is incoherent in explaining the mechanism. Second, the color marked on the cross-sectional STEM image is not necessary. It seems like there is another "something" (another layer?) in the area marked Al₂O₃.

(7) No further comment.

One more question,

Compared to Fig. 3h, some of the TEM images in Supplementary Fig. 8 don't show 3R stacking.

Reply to Referee #2

Original comment (1):

Authors have revised the manuscript according to the comments presented by two reviewers and replied all my questions. I would like to suggest that the manuscript may be accepted after comparing the growth mechanism in this manuscript with that of review article on *Nanoscale* (2024, 16, 978, Large-area single-crystal TMD growth modulated by sapphire substrates).

Our reply:

We greatly thank the referee's valuable suggestion to compare our growth mechanism with that described in the review article on *Nanoscale* (Large-area single-crystal TMD growth modulated by sapphire substrates). This work divides the growth mechanism of single-crystal TMD into van der Waals epitaxy, step-guided epitaxy, and dual-coupling guided epitaxy.

The growth mechanism in our manuscript is one of the dual-coupling guided epitaxy, and the TMD layers are not directly bonded to the steps. So, they can be grown across the steps rather than only along the steps, which is consistent with our previous works (*Nature Nanotechnology* 2022, 17, 33; *Nature Communications* 2023, 14, 592).

We have added these discussions and cited the suggested reference in the revised manuscript.

Reply to Referee #3

Many thanks to the authors for the response.

My comments to authors' response,

Original comment (1):

(2) No further comment.

(3) No further comment.

(4) No Further comment.

Original comment (2):

The SHG mapping images (Supplementary Fig. 11) should include intensity scale bar, and 3R stacking should have stronger SHG intensity, but from those images it's hard to tell; otherwise, those images seem like a monolayer, noise or not on focus.

Our reply:

We appreciate the referee's suggestion on adding the intensity scale bar of the SHG mapping. We agree with the referee that the 3R stacking should have stronger SHG intensity. To make this clearer, we have added the SHG mapping of monolayer WS₂ film under the same experimental settings (Fig. R1).

We have added this data in the revised manuscript.

Fig. R1. a, SHG mapping of a monolayer WS₂ sample. b-p, SHG mapping of bilayer WS₂ samples at different areas. No dark lines can be observed, demonstrating the seamless stitching of different WS₂ islands.

Original comment (3):

The orientation of the triangular WS₂ in Supplementary Fig. 4 (with the edge of the triangle parallel to the step edge) is different from the triangle in Supplementary Fig. 6 (the edge of the triangle perpendicular to the step edge), which is incoherent in explaining the mechanism. Second, the color marked on the cross-sectional STEM image is not necessary. It seems like there is another "something" (another layer?) in the area marked Al₂O₃.

Our reply:

We greatly thank the referee for highlighting the different orientations of WS₂ in Supplementary Fig. 4 and 6. In fact, they are not contradictory as the WS₂ island in Supplementary Fig. 4 is a monolayer one, and the WS₂ islands in Supplementary Fig. 6 are bilayer ones. Supplementary Fig. 4 shows that the morphology of the steps on a sapphire substrate can be faithfully replicated to the as-grown monolayer WS₂. And Supplementary Fig. 6 demonstrated the aligned growth of bilayer WS₂ islands near the steps edges.

For the cross-sectional STEM image, the feature noted as "another layer" in the area marked Al₂O₃ is not an additional layer but likely a surface reconstruction of Al₂O₃. This phenomenon has also been observed in previous works (marked with red box in Fig. R2, Nature Nanotechnology 2023, 18, 448).

We have deleted the colour in the TEM image and revised the Supplementary Fig. 7 in the revised manuscript.

Fig. R2. Cross-sectional STEM image of a multilayer SnS₂/monolayer WS₂/sapphire structure in Nature Nanotechnology 2023, 18, 448.

Original comment (4):

(7)No further comment.

Original comment (5):

Compared to Fig. 3h, some of the TEM images in Supplementary Fig. 8 don't show 3R stacking.

Our reply:

We greatly thank the referee's concern on the TEM images in Supplementary Fig. 8. Variations in electron beam dosage, as well as brightness and contrast settings, can indeed lead to differences in contrast across various regions of the samples. Nonetheless, it is definitive that our samples exhibit 3R stacking of WS_2 , as confirmed by the line profile intensities.

Specifically, Fig. R3a (Fig. 3h) and Fig. R3b (Fig. S8) display identical line profile intensities, each presenting three distinct intensity levels corresponding to W, S2, and W+S2. This tripartite intensity profile is a hallmark of 3R stacking. In contrast, Figure R1c illustrates the characteristics of 2H stacking, which is marked by a single intensity type corresponding to W+S2.

To make the 3R stacking clearer, we have tuned the brightness and contrast of the TEM images in Supplementary Fig. 8.

Fig. R3. STEM images of 3R (a-b) and 2H (c) stacked WS_2 .